# Isolation and Characterization of the *Acadevirus* Members BigMira and MidiMira Infecting a Highly Pathogenic *Proteus mirabilis* Strain

**DOI:** 10.3390/microorganisms11092141

**Published:** 2023-08-23

**Authors:** Jéssica Duarte da Silva, Lene Bens, Adriele J. do Carmo Santos, Rob Lavigne, José Soares, Luís D. R. Melo, Marta Vallino, Roberto Sousa Dias, Zuzanna Drulis-Kawa, Sérgio Oliveira de Paula, Jeroen Wagemans

**Affiliations:** 1Laboratory of Molecular Immunovirology, Department of Microbiology, Federal University of Viçosa, Viçosa 36570-900, MG, Brazil; jessica.duart.s@gmail.com (J.D.d.S.); adriele.docarmo27@gmail.com (A.J.d.C.S.); jose.j.soares@ufv.br (J.S.); depaula@ufv.br (S.O.d.P.); 2Laboratory of Gene Technology, Department of Biosystems, Division of Animal and Human Health Engineering, KU Leuven, 3000 Leuven, Belgium; lene.bens@kuleuven.be (L.B.); rob.lavigne@kuleuven.be (R.L.); 3Centre of Biological Engineering, University of Minho, 4710-057 Braga, Portugal; lmelo@deb.uminho.pt; 4LABBELS—Associate Laboratory, 4710-057 Braga, Portugal; 5Institute for Sustainable Plant Protection, National Research Council of Italy, 10135 Torino, Italy; marta.vallino@ipsp.cnr.it; 6Department of General Biology, Federal University of Viçosa, Viçosa 36570-900, MG, Brazil; rosousa318@gmail.com; 7Department of Pathogen Biology and Immunology, University of Wroclaw, 50-335 Wroclaw, Poland; zuzanna.drulis-kawa@uwr.edu.pl

**Keywords:** podoviruses, *Acadevirus*, depolymerases, CAUTI, *Proteus mirabilis*

## Abstract

*Proteus mirabilis* is an opportunistic pathogen and is responsible for more than 40% of all cases of catheter-associated urinary tract infections (CAUTIs). Healthcare-associated infections have been aggravated by the constant emergence of antibiotic-resistant bacterial strains. Because of this, the use of phages to combat bacterial infections gained renewed interest. In this study, we describe the biological and genomic features of two *P. mirabilis* phages, named BigMira and MidiMira. These phages belong to the *Acadevirus* genus (family *Autographiviridae*). BigMira and MidiMira are highly similar, differing only in four missense mutations in their phage tail fiber. These mutations are sufficient to impact the phages’ depolymerase activity. Subsequently, the comparative genomic analysis of ten clinical *P. mirabilis* strains revealed differences in their antibiotic resistance profiles and lipopolysaccharide locus, with the latter potentially explaining the host range data of the phages. The massive presence of antimicrobial resistance genes, especially in the phages’ isolation strain *P. mirabilis* MCS, highlights the challenges in treating infections caused by multidrug-resistant bacteria. The findings reinforce BigMira and MidiMira phages as candidates for phage therapy purposes.

## 1. Introduction

*Proteus mirabilis* is a Gram-negative opportunistic pathogen, well known as one of the major causes of catheter-associated urinary tract infections (CAUTIs), and a common cause of secondary bloodstream and healthcare-related infections [1,2,3]. *P. mirabilis* is also routinely isolated from extra-intestinal infections, such as wounds; eye, nose, and skin meningoencephalitis; and osteomyelitis [4,5,6]. The pathogenicity of this bacterium is associated with its remarkable swarming capacity, combined with the production of adhesive virulence factors, pili, and fimbriae, resulting in the formation of robust translucent biofilms in catheter devices (which can cause encrustations and even block the flow of urine). Moreover, *P. mirabilis* produces ureases that hydrolyze urea into ammonia and carbon dioxide, a process that favors the development of infection-induced stones [1,6,7]. More and more *P. mirabilis* strains resistant to antibiotics commonly used to treat urinary tract infections, such as fluoroquinolones, aminoglycosides, and even extended-spectrum β-lactamases, are being isolated, thereby raising concerns about infections caused by this organism [3,4,5,6,7].

The use of phages, viruses that infect bacteria, to combat bacterial infections has been widely explored since their discovery at the beginning of the 20th century [8,9]. This strategy, called phage therapy, began with great success but lost its appeal due to inconsistencies in treatments (mainly related to the lack of knowledge about viral infectious mechanisms) and with the establishment of antibiotics as a cheap and effective treatment [9,10,11]. However, the evolutionary pressure of antibiotics, combined with their overuse, has resulted in an alarming rise of antibiotic-resistant strains, which have become a worldwide source of concern and are leading to an antimicrobial resistance (AMR) crisis [12,13,14]. The COVID-19 pandemic has elevated this problem to a new level, as evidenced by the last reports of the US Center for Disease Control and Prevention (CDC) and the Pan American Health Organization (PAHO) [14,15]. These reports found an alarming rise in the incidence of Gram-negative AMR bacteria, including carbapenem-resistant *Acinetobacter*, extended-spectrum beta-lactamase (ESBL)-producing Enterobacterales, and carbapenem-resistant Enterobacterales, and describe a high level of antibiotic prescription to patients with COVID-19, despite the relatively low proportion of patients who actually developed secondary bacterial infections. The latter is thought to be primarily responsible for an increase of 15% in mortality and resistant hospital-onset infections in 2020 [14,15,16,17]. Agencies such as the World Health Organization (WHO) invest significant resources in initiatives that aim to bring solutions to the antimicrobial resistance crisis [13,16], and phage therapy is considered again an alternative for the treatment of bacterial infections.

The use of lytic phages to treat bacterial infections has various advantages, including host specificity, low toxicity, fast isolation of specific phages, potentially low production costs, and the irremediable death of the host at the end of the infection cycle [18,19,20]. Despite this, the general narrow host range and the possible emergence of resistance to phages by the bacterial population are challenges to be overcome. A well-known approach to addressing these problems is the use of phage cocktails, which can infect multiple bacterial strains and decrease the development of resistance mechanisms [21,22,23].

Although countries like Belgium, Georgia, Poland, and Russia already have specific regulations for phage therapy, and others like the United Kingdom, France, and the United States are moving in this direction, the lack of laws that regulate the use of phages around the world is also a factor that hinders the wide spread of this approach, especially when the objective is recurrent use and not a therapy of last resort [24,25,26]. For this reason, the global phage scientific community is committed to the establishment of high-quality clinical trials, such as PhagoBurn, or multidisciplinary approaches, such as the PHAGEFORCE study, to create solid and accurate data about the safety and efficacy of phage therapy [27,28,29]. Systematic reviews are additionally being developed in order to organize and assign statistical significance to previous non-standardized clinical studies and case reports that have been published [30,31]. Because of this, new studies demonstrating the efficacy of phage therapy, the isolation and characterization of novel phages, and, consequently, the advancement of knowledge of these viruses are essential steps for phage therapy to be considered as a safe and beneficial approach for treating bacterial infections.

This work aimed to characterize biologically and genomically the *Proteus* phages BigMira-UFV01 and MidiMira-UFV02, isolated against the super-resistant clinical strain *Proteus mirabilis* MCS, and to assess their potential as phage therapy agents.

## 2. Materials and Methods

### 2.1. Phage Isolation, Propagation, and Purification

Phages BigMira-UFV01 and MidiMira-UFV02 (called BigMira and MidiMira) were isolated from aqueous samples from a swine farmer located in the city of Viçosa (Minas Gerais, Brazil) using a protocol adapted from Van Twest and Kropinski (2009) [32]. Briefly, the liquid sample was repeatedly centrifuged at 10,000× *g* for 15 min, until no visible particles remained in the supernatant, and then filtered through 0.45 μm and 0.22 μm filters. Then, 10 mL of the filtered supernatant was added to the same volume of 2X lysogeny broth (LB) medium. Isolation host *P. mirabilis* MCS, isolated from a chronic wound of a diabetic patient, was grown to the exponential growth phase and added to the mixture (100 µL), followed by overnight incubation at 37 °C, while shaking (100 rpm). After the incubation, a double agar overlay assay was performed [32]. The lysis plaques resulting from this process that showed a distinct morphology were picked from the agar and propagated independently in LB medium containing the host. This process was repeated at least five times. When phages were considered pure, they were concentrated and purified by PEG precipitation [33]. Hence, 20 mL of a solution of 25% (*p*/*v*) PEG8000 was added to 30 mL of the previously filtered and pure phage supernatant, reaching a final concentration of 10% PEG. The mixture was kept at 4 °C overnight, under agitation of 100 rpm, and then centrifuged at 12,000× *g* for 30 min. The supernatant was discarded, and the pellet was resuspended in 400 μL of phage buffer.

### 2.2. Biological Features

#### 2.2.1. Lysis Plaque Measurements

The diameters of fifteen lysis plaques resulting from the BigMira infections and fifteen lysis plaques resulting from the MidiMira infections were measured using the software ImageJ (https://imagej.net/ij/ (accessed on 19 August 2023)) [34].

#### 2.2.2. Transmission Electron Microscopy (TEM)

TEM micrographs were obtained following the protocol described by Vallino et al. (2021) [35]. Briefly, 10 µL of purified phage stock (about 10^9^ PFU/mL) was dropped onto a carbon/Formvar-coated grid and set aside for three minutes. Uranyl acetate (0.5% *w*/*v*) was used for negative staining. Observations and image acquisition were performed using an 80 kV CM 10 electron microscope (Philips, Eindhoven, The Netherlands).

#### 2.2.3. Host Range

The host range of BigMira and MidiMira was determined both by spot and killing assays. Briefly, a panel of clinical *P. mirabilis* strains (Table 1) was incubated overnight in LB medium, without agitation, at 37 °C. An aliquot of 700 µL of grown bacteria was mixed with 5 mL of LB top agar (0.7%) and poured into a Petri dish containing LB bottom agar (1.5%). After solidification, 5 μL of the phage stock (about 5 × 10^6^ PFU/mL) was dropped on the bacterial lawn. The absence of bacterial growth where the phage suspensions were dropped confirmed that the bacterial strain was a phage host. By way of comparison, the *Proteus* phage vB_PmiP_Pm5460 [36] was also tested against the clinical *P. mirabilis* panel. The phage-killing curves were constructed by taking consecutive OD_600_ absorbance measurements every 15 min, for 24 h. Briefly, 10 μL of BigMira and MidiMira (final concentration of 10^8^ PFU/mL) was added to 190 μL of freshly grown *P. mirabilis* strains (OD_600_ of 0.1) in a 96-well plate. The experiment was performed in triplicate and repeated in three biological replicates. The bacterial growth curves (with and without phages) were compared to identify differences between the growth curves.

#### 2.2.4. One-Step Growth Curve

A one-step growth curve was performed to determine the infection behavior (latency time and burst size) of BigMira and MidiMira. Phages were added to 4 mL of the host *P. mirabilis* MCS, in the early exponential phase (OD_600_ = 0.4; about 10^8^ CFU/mL) at a final concentration of 10^3^ PFU/mL, to obtain a multiplicity of infection (MOI) of 0.00001. A first sample (100 μL) was collected to determine the titer of phages at the beginning of the experiment. The phage/host mixture was incubated for five minutes, at 37 °C and then centrifuged for eight minutes at 6000× *g*. The supernatant was discarded together with the unabsorbed phages, after which the pellet was gently washed and resuspended in 5 mL of LB. Samples were collected in intervals of 5 or 10 min over one hour, and the phage titer was immediately quantified by double-layer agar assay. The experiment was performed in three biological replicates. The burst size was calculated as follows: burst size = average of first phage peak (PFU/mL)/average of the initial phage titer.

#### 2.2.5. Phage Stability

The stability of the BigMira and MidiMira phages was measured under different conditions. For thermal stability, five temperatures were evaluated (−80, −20, 4, 37, and 55 °C). The phages (final concentration of 10^8^ PFU/mL) were diluted in phage buffer (10 mM Tris. HCl; 10 mM MgSO_4_; 150 mM NaCl; pH 7.5) and kept for 48 h at the tested temperatures. For pH stability, several pH values were tested (3, 4, 6, 7, 9, 11, 12, and 13). The phages were diluted in pH buffer (150 mM KCl; 10 mM KH_2_PO_4_; 10 mM Na citrate; 10 mM H_3_BO_3_), with adjusted pH values, and kept in triplicate for 48 h at 25 °C. To evaluate the phages’ stability in an environment similar to a real UTI, the phages (final concentration of 10^8^ PFU/mL) were incubated at 37 °C in voided and sterile (filtered in 0.22 µm) urine. The phages were titered after 24 and 48 h, using a spot assay. The phages’ propagation capacity was also tested in this condition. The phages (final concentration of 10^4^ PFU/mL) and their host *P. mirabilis* MCS (final concentration of 10^8^ CFU/mL) were kept at 37 °C in voided and sterile urine. The titration was performed using a spot assay after 24 h. The experiments were performed in triplicate and repeated in three biological replicates. An ANOVA two-way analysis was used to compare the average differences among the treatments (*p*-value 0.05).

### 2.3. Genome Analysis

#### 2.3.1. DNA Extraction and Sequencing

The phages’ DNA was extracted using the protocol described by Kot (2018) [37]. Briefly, in a 1.5 mL tube, 90 μL of phage lysate was filtered through a 0.45 μm ultrafiltration spin-column and then mixed with 10 μL of DNase I buffer and 5 U of DNase I. After 30 min of incubation (37 °C), 10 μL of 50 mM EDTA and 10 μL of 1% SDS were added for DNase I inactivation. Then, 5 μL of proteinase K was added, and the mixture was kept at 55 °C for 45 min. The viral DNA was then purified using the DNA Clean and Concentrator kit. The sequencing was performed on an Illumina MiniSeq device (San Diego, CA, USA) (2*150 bp paired reads) with a library generated with the Nextera Flex DNA library kit (Illumina).

#### 2.3.2. Assembly and Annotation

The raw sequences of BigMira and MidiMira were assembled using default parameters of the “Assembly tool” on the Bacterial and Viral Bioinformatics Resource Center (PATRIC) (https://www.bv-brc.org/ (accessed on 19 August 2023)) [38]. The resulting contigs were annotated using the “annotation tool” on PATRIC, following the “Classic RAST pipeline” (Rapid Annotation using Subsystem Technology) (https://rast.nmpdr.org/rast.cgi (accessed on 19 August 2023)) [39], and using the PROKKA database [40]. The ORFs were manually checked, and the consensus CDSs were maintained on the fasta and GenBank files. Host promoters were predicted using Sapphire (https://sapphire.biw.kuleuven.be/ (accessed on 19 August 2023)) [41], and phage promoters were predicted using Multiple Em for Motif Elicitation (MEME) (https://meme-suite.org/meme/tools/meme (accessed on 19 August 2023)) [42], followed by manual checking. tRNAscan_SE [43] was used to search for tRNAs, and ARNold was used to find terminators for the identification of Rho-independent terminators [44].

#### 2.3.3. Genomic and Phylogenetic Analysis

The Viral Proteomic Tree server (Viptree) (https://www.genome.jp/viptree/ (accessed on 19 August 2023)) [45] was used to identify the proteomic similarity between BigMira and MidiMira and the reference genomes from its database. To identify the similarity of both phages with the National Center for Biotechnology Information (NCBI) database (https://blast.ncbi.nlm.nih.gov/Blast (accessed on 19 August 2023)), a comparative Megablast was performed. The start of the genomes of BigMira and MidiMira and their relatives was chosen using *Proteus* phage vB_PmiP_Pm5460 as a reference. The pairwise intergenomic distances/similarities of the genomes were calculated using VIRIDIC (http://rhea.icbm.uni-oldenburg.de/VIRIDIC/ (accessed on 19 August 2023)) [46] and then aligned using Clinker [47] to generate a gene cluster comparison. The Snippy tool from Galaxy Australia (https://usegalaxy.org.au/ (accessed on 19 August 2023)) was used to calculate putative SNPs between the BigMira and MidiMira genomes.

#### 2.3.4. Putative Depolymerase Enzyme Search and Tertiary Structure Prediction

The search for depolymerase-like enzymes was performed using the tool Phage Depolymerase Finder [48] (Galaxy Version 0.1.0) from the Galaxy Docker Build platform (https://galaxy.bio.di.uminho.pt/ (accessed on 19 August 2023)). The proteins predicted as putative depolymerases were then submitted for sensitive sequence searching based on profile HMMs (HMMER) [49], and homology detection and structure prediction by HMM–HMM comparison (HHPRED) [50] on the MPI bioinformatics toolkit (https://toolkit.tuebingen.mpg.de/ (accessed on 19 August 2023)) [51] for an accurate search for conserved domains. The AlphaFold2 [52] pipeline was used to predict the tertiary structure of the proteins, using version 1.3.0 and the default settings.

### 2.4. Proteus mirabilis Clinical Strains

#### 2.4.1. Bacterial Strains

*Proteus mirabilis* MCS was isolated from a pressure ulcer wound of a diabetic female patient in Brazil, who was also suffering from a chronic case of urinary tract infection. The ulcer was located on the sacral region of the patient’s back and developed during a long period of hospitalization due to a severe case of COVID-19. The *P. mirabilis* clinical strains were isolated at university hospitals in Leuven (Belgium) from patients of both genders suffering from hidradenitis suppurativa, a skin disease. *P. mirabilis* SGSC 5460 [36] was also used in this study in the host range assay. The bacterial isolates were grown in LB media at 37 °C.

#### 2.4.2. *Proteus mirabilis* DNA Extraction and Sequencing

The bacterial genomes were extracted using the DNeasy UltraClean Microbial Kit Handbook (Qiagen, Hilden, Germany), following the protocol instructions. Illumina sequencing was performed as described for the phage genomes.

#### 2.4.3. Genome Assembly and Annotation

The raw sequencing data of the *P. mirabilis* clinical strains were assembled using the default parameters of the “Assembly tool” on PATRIC (https://www.bv-brc.org/ (accessed on 19 August 2023)) [38]. The resulting contigs were annotated using RAST and the PROKKA database.

#### 2.4.4. Bioinformatics Analysis

For the core genome determination, Roary [53] was run. The Core Gene Alignment files were then submitted to RAxML—maximum likelihood-based inference of large phylogenetic trees [54]. Both tools are available on the Galaxy Australia platform (https://usegalaxy.org.au/ (accessed on 19 August 2023)). The core genome alignment visualization was created using Phandango—Interactive visualization of genome phylogenies (https://jameshadfield.github.io/phandango/#/ (accessed on 19 August 2023)) [55]. The lipopolysaccharide (LPS) locus of the *P. mirabilis* clinical strains was predicted by the Subsystem Features Categories on Rast SEED Viewer (https://rast.nmpdr.org/rast.cgi (accessed on 19 August 2023)) [39], using the “Cell Wall and Capsule” category, “Gram-Negative cell wall components” subcategory, and “Lipopolysaccharide assembly” subsystem. The genes related to the LPS locus were then organized in GFF3 files and submitted on the same pipeline previously described. Finally, the analysis tool Resistance Gene Identifier (RGI) on the Comprehensive Antibiotic Resistance Database (CARD) was used to perform in silico detection of antibiotic resistance genes among the isolates used in this study.

## 3. Results

### 3.1. Phage Isolation

Two phages were isolated against the host *Proteus mirabilis* MCS, using swine farm samples. Based on the different morphologies of the lysis plaques (Figure 1A,B), the phages were named BigMira-UFV01 (BigMira), which presents bigger lysis plaques (average diameter of 26.85 ± 1.81 mm) surrounded by halos (99.83 ± 6.9 mm), and MidiMira-UFV02 (MidiMira), which possesses smaller (average diameter of 12.66 ± 1.5 mm) but still clear lysis plaques without halos.

### 3.2. Biological Features

The micrographs obtained by TEM showed that both phages BigMira (Figure 1C) and MidiMira (Figure 1D) possess the morphology typical of podoviruses, characterized by an icosahedral capsid and a short tail. Also, as confirmed both by spot and killing assays, BigMira and MidiMira were only able to infect their specific isolation host strain MCS. The results are summarized in Table 1.

A one-step growth curve analysis (Figure 2A,B) shows that both BigMira and MidiMira have a latent period of approximately 15 min. The estimated burst size was 13 phage particles per infected cell (p.p/i.c) for BigMira and 39 p.p/i.c for MidiMira.

The phages’ thermal stability was measured after 24 and 48 h of treatment at temperatures of −80, −20, 4, 37, and 55 °C (Figure 3). For both phages, no viral particles were detected after 24 h incubation at 55 °C. The incubation time (24 or 48 h) at remaining temperatures did not alter the viral stability under the tested conditions. Also, BigMira showed a discrete (one order decrease) yet significant (*p*-value ≤ 0.05) viral titer alteration at −80, −20, and 4 °C when compared to the 37 °C sample.

The phages’ stability at different pH values was measured after 24 and 48 h of incubation. BigMira and MidiMira did not differ from each other in stability. In both cases, no significant variations in viral titer were found when the phages were incubated at a pH of 6, 7, or 9. On the other hand, no viral particles were detected when the phages were kept at pH values of 3, 11, 12, and 13. Furthermore, the only scenario where the incubation time showed a significant difference (*p*-value 0.05) was at pH 4, where fewer phage particles were detected after 48 h of incubation than after 24 h. In terms of stability in urine, the results indicate that the phages maintained their titers even after 48 h of incubation at 37 °C and the fact that the urine was sterile or voided had no effect on the viral stability. The phages were also able to infect their host *P. mirabilis* MCS and propagate normally in urine (Appendix A).

### 3.3. Genomic Features

The BigMira and MidiMira sequencing results revealed that both phages have a dsDNA genome of 43,026 bp with a GC content of 39.4%. They contain 52 open reading frames (ORFs) and no predicted tRNAs. None of the predicted ORFs encode lysogeny-associated proteins, allowing the classification of these phages as virulent.

A search for related phages using BLASTn showed that BigMira and MidiMira present more than 95% identity with *Proteus* phages PM 116 (NC_047858), PM 93 (NC_027390), PM 85 (NC_027379), and vB_PmiP_Pm5460 (NC_28916). All these phages belong to the same *Acadevirus* genus within the *Molineuxvirinae* subfamily and *Autographiviridae* family. The *Citrobacter* phage vB_CroP_CrRp3 (NC_047920), the next closest neighbor, only showed 75.65% identity and belongs to another genus, *Vectrevirus*. A proteome analysis using the Viptree database confirmed that the *Proteus* phages form a distinct clade (Figure 4A).

The intergenomic distance between the related phages calculated by VIRIDIC indicates that BigMira and MidiMira share 99% similarity and belong to the same species. The phages also share more than 70% intergenomic similarity with PM 116, PM 93, PM 85, and Pm5460, indicating that they are members of the same genus, but of a different species (Figure 4B). The main features of the different phages within the genus *Acadevirus* are summarized in Table 2. All of them had been isolated from *Proteus mirabilis* strains and have similar G+C content and genome length. Other biological characteristics, such as their burst sizes, exhibit higher variances. As shown in Table 1, BigMira and MidiMira were also evaluated for their ability to infect the host strain of *Acadevirus* Pm5460, namely *P. mirabilis* 5460. However, it could not infect this strain. Phage Pm5460, on the other hand, could also not infect *P. mirabilis* MCS, the host strain of BigMira and MidiMira, indicating a narrow host range for these types of viruses, which was also observed for PM 85, PM 93, and PM 116 (Table 2).

An alignment of BigMira and MidiMira and their relatives (Figure 5B) demonstrates the similarity between their genome architectures. The early gene module, although relatively well conserved within the genus, appears to be unique for *Acadevirus* phages. A BLASTn analysis showed that these early genes, particularly those encoding hypothetical proteins, had no similarity with phages from other genera. The general genome organization of these phages is typical for the *Autographiviridae* family, and except for the last two proteins, the ORFs are highly conserved. Due to the extensive sequence similarity between BigMira and MidiMira, only the BigMira genomic map is depicted in Figure 5A.

### 3.4. Putative Depolymerase-Encoding Domain Prediction

A search for proteins with biotechnological potential revealed that all the published *Acadevirus* phages contain putative depolymerase domains. Their genes are located in the least conserved region of the *Acadevirus* genomes. In the case of *Proteus* phages PM 116 and PM 85, only the phage tail fiber is predicted to possess depolymerase activity. For BigMira, MidiMira, Pm5460, and PM 93, the predicted domains can be found in two proteins, the phage tail fiber (gp51) and the hypothetical protein located immediately downstream (gp52). A promoter prediction shows that the promoters on the BigMira/MidiMira genomes are located at the same sites as those in *Proteus* phage vB_PmiP_Pm5460 and that in both cases, the expression of the proteins with depolymerase domains is controlled by an individual promoter. The Rho-independent terminators are also located in the same regions.

Given the high similarity between the viral genomes (same length, ORFs, and organization), but a different plaque morphology, a single-nucleotide polymorphism (SNP) analysis was performed to search for mutations that could explain the differences between the phages. The results are summarized in Table 3.

Only four SNPs leading to a non-synonymous mutation in the protein were observed, all present in the same gene, encoding the phage tail fiber Gp51, one of the predicted proteins containing a depolymerase domain (Table 3). Figure 6 displays a tertiary structure prediction of the BigMira and MidiMira phage tail fiber Gp51. The colored areas show the sites where the amino acids were changed. The substitution of a tyrosine (BigMira) for a cysteine (MidiMira) in the central monomer’s beta-helical region is colored in blue. The substitution of a glycine for a serine and an aspartic acid for an asparagine happened just with six nucleotides of difference and are colored in pink and red, respectively. Finally, the replacement of phenylalanine for leucine is colored yellow. These substitutions are located on the C-terminal region of the protein. AlphaFold2 did not support the prediction of a protein trimer, often found on phage tail fibers. Hence, the prediction of the modified regions in a model closer to the phage physiological reality was not possible.

BLASTp, HHPRED, and HMMER were used to search for conserved domains or distant homologies that could assign a function to the hypothetical protein Gp52. While no similarities were found using BLASTp or HMMER, the HHPRED results (more than 50 hits with high similarity) indicated that the protein may be classified as a tail spike with a hydrolase or lyase activity. AlphaFold2 was used to predict the 3D structure of a Gp52 monomer (Appendix A) and trimer (Appendix A). However, due to the lack of similarity of this protein with other proteins in the AlphaFold database, although the prediction suggested a common depolymerase RBP structure, the resulting prediction had low quality and failed to present the correct folding of several regions of the protein, including its catalytic site.

### 3.5. Proteus mirabilis Clinical Strain Genomic Analysis

The pan-genome analysis of all used *P. mirabilis* clinical strains is shown in Figure 7A. The genome profile of the phages’ host *P. mirabilis* MCS is most similar to that of the Belgian strain *P. mirabilis* 082. Next, we investigated the LPS locus, since LPS is most likely the primary receptor of the acadeviruses’ receptor-binding protein (RBP), with its enzymatic activity enabling the start of the infection process. Interestingly, when only considering the LPS locus (Figure 7B), *P. mirabilis* MCS does not cluster with other clinical isolates, demonstrating that the LPS composition of this strain is unique and may be significantly different from the others, thereby potentially explaining the observed narrow host range.

Still aiming to explore the differences and similarities among the clinical strains of *P. mirabilis*, the antibiotic resistance genes of each isolate were identified (Table 4). The genes *kpnH*, *gyrB*, *rsmA*, *catA4*, and *crp* are considered core genes that are found in all isolates, except for *crp*, which is missing in *P. mirabilis* 218. *kpnH*, *rsmA*, and *crp* are related to multidrug efflux pumps, while *gyrB* is important for the quinolone resistance mechanism and *catA4* is important for chloramphenicol resistance. The strains *P. mirabilis* 074, *P. mirabilis* 082, *P. mirabilis* 129, and *P. mirabilis* 218 possess just the genes *qnrD1*, *tetQ*, and *bla*_TEM-2_ in addition to the core ones, respectively. *P. mirabilis* 114, *P. mirabilis* 159, and *P. mirabilis* 163 present the same resistance profile and share the genes *vat* and *dfrA1*. Beyond the previous genes, *P. mirabilis* 195 also possesses the genes *aadA* and *catII*.

The strain *P. mirabilis* 204 possesses more than eight genes on top of the core resistome that confer resistance against four different classes of antibiotics: aminoglycosides—*aadA*, *aadA2*, *aph(3′)-Ia*, *aph(6)-Id aph(3″)-lb*; tetracyclines—*tetA*; β-lactams—*bla*_TEM-135_; and sulfonamides—*sulI*, presenting a high antimicrobial resistance profile.

The most remarkable antimicrobial resistance profile among the evaluated isolates was found for *P. mirabilis* MCS, the host strain of BigMira and MidiMira, with 17 resistance genes being identified. Five are considered core genes, and the others confer resistance to eight distinct classes of antibiotics: tetracyclines—*tetA*, *tetQ*; aminoglycosides—*aac(6′)-Iq*, *aac(6′)-Ib′*, *aadA*; diaminopyramidines—*dfrA1*; β-lactams—*bla*_OXA-9_, *bla*_CTX-M-2_; streptogramins—*vat*; chloramphenicol—*catA2*; sulfonamides—*sul1*; and the gene *qacEdeltal* which confers resistance to disinfecting agents and antiseptics, such as ethidium bromide.

## 4. Discussion

Catheter-associated urinary tract infections (CAUTIs) are a common and serious healthcare-associated issue. Among the microorganisms that cause CAUTIs, *Proteus mirabilis* is one of the major pathogens, responsible for up to 40% of all cases [2,6,7,36]. *P. mirabilis* is known for its ability to form crystalline biofilms in catheter devices, leading to encrustations and blockage of the flow of urine, and for producing ureases that can promote the formation of stones in the urinary tract, cystitis, pyelonephritis, and difficult-to-treat infections [1,6,57,58]. The emergence of *P. mirabilis* strains resistant to different antibiotic classes worsens the problems caused by this organism. Unfortunately, the acquisition of antibiotic resistance genes has not only occurred with *P. mirabilis*, and over the years, this problem has become more unmanageable. The enormous number of hospitalizations and antibiotics ingested during the COVID-19 pandemic accelerated this problem, especially for Gram-negative bacteria [14,15,17].

This study describes the isolation and characterization of two *P. mirabilis* phages, named BigMira and MidiMira. The phages were isolated from swine farm samples, and their plaque morphology was the main criterion used to separate them. TEM revealed that both phages present a podovirus morphology, and phylogenetic analysis confirmed that they belong to the family *Autographiviridae* and the genus *Acadevirus*. Only four other phage genomes of this genus are available on NCBI: PM116, PM85, PM93 [56], and vB_PmiP_Pm5460 [36]. These genomes are also the only ones found in the taxonomic browser of the International Committee on Taxonomy of Viruses and in the Viptree reference genome database, implying that few phages from this genus have been isolated thus far and that *Acadevirus* is a small group with highly conserved characteristics. Until 2019, *Acadevirus* members were classified as T7-like phages. After that, the family *Autographiviridae* became an independent family, containing 9 subfamilies and 52 genera, including *Acadevirus*. The genus is named after the district of Academgorodok, part of the Russian city of Novosibirsk, from which the type species *Proteus* phage PM85 was isolated [59]. However, despite the low nucleotide similarity between acadeviruses and the more prevalent teseptimaviruses (Figure 4B), they share a similar gene architecture and a distinct separation between early, middle, and late genes [60].

Morozova et al. tested 37 hosts and described the *Proteus* phages PM 85, PM 93, and PM 116 as having a narrow host range, once they were able to infect just 1–3 *P. mirabilis* strains [56]. On the other hand, Melo et al. described *Proteus* phage Pm5460 as capable of infecting almost 62% of the tested 24 *Proteus* spp. strains [36]. Although BigMira and MidiMira appear to have a narrow host range (only infecting the isolation host) (Table 1), they were tested against a smaller host strain collection compared to their relatives and were not tested against hosts from other species of the same origin as strain MCS (Table 2). Thus, further investigation is required before the phage’s narrow host range can be confirmed. The one-step growth curve revealed that the *Acadevirus* phages have a latent period that ranges from 7 to 15 min and an average burst size of 43.5 phage particles per infected cell (p.p/i.c) (Table 2) [36,56]. It is important to point out that the burst size can be calculated in different ways, but we chose to use the same formula as the previous studies to enable comparison.

BigMira and MidiMira also showed similar results on the stability tests. Both phages were not able to endure a temperature of 55 °C but were stable in the other tested temperatures. Regarding pH stability, the phages do not withstand pH 3, 11, 12, and 13. pH 4 seems to be the threshold for these phages since they slowly lost stability over the 48 h of the experiment. This was the only condition where the incubation time interfered with viral stability. Even though these are the first data on *Acadevirus* stability (the previous studies did not include these trials), the findings are comparable to those obtained for other podoviruses. In general, these phages are resistant to low temperatures (including long-term storage at −80 and −20 °C) and are stable at neutral and mildly basic or acidic pHs. Temperatures above 55 °C or pH values below 4 and above 10 usually render these virus particles inactive [61,62,63,64]. Both phages also proved to be stable in environments that mimic the conditions of urinary tract infections, and their titers remained unaltered even when incubated in voided and sterile urine for 48 h at 37 °C. They also maintained their ability to infect and propagate in their host unaltered under these conditions. The fact that the phages BigMira and MidiMira maintained their titers under different thermal, pH, and urine conditions reinforces that they can be kept for up to 48 h in non-refrigerated situations and further strengthens their candidacy as phage therapy agents.

The genomic features of BigMira and MidiMira also resemble those described for other phages of the genus *Acadevirus* (Table 2). The genome size of 43,026 bp, the G+C content of 39.4%, and the 52 predicted coding sequences (CDSs) are consistent with the genus averages of 44,006 bp, 39.38% GC content, and 51.3 ORFs, respectively [36,56]. The differences in the number of CDSs for each phage can be attributable primarily to small hypothetical proteins found in different regions of the genome. These proteins affect genomic alignment and appear to be poorly conserved since they have a low rate of similarity with other proteins found in databases such as NCBI. The active search to assign functions to these hypothetical proteins in programs such as HHPred and HMMR also ends up leading to poorly conserved domains and functions that do not fit bacteriophages. Therefore, the “hypothetical protein” annotation, with no function attribution, was maintained.

A feature observed in all the acadeviruses described until now is the presence of a translucent halo around the lysis plaques resulting from their infective process [36,56]. This halo is characteristic of the presence of depolymerase-type enzymes in the phage tail fibers [65]. The prediction of putative depolymerases (DPOs) indicated that in the case of BigMira and MidiMira, two ORFs have depolymerase-like domains, the phage tail fiber Gp51 and hypothetical protein Gp52. As shown in Figure 5, these genes are found in the least conserved region of the genome of the acadeviruses; some phages present just one putative depolymerase, and the others possess two [36,56]. Latka et al. described the architecture of depolymerase-containing receptor-binding proteins (RBPs) [66]. These proteins are usually annotated as tail fibers, tail spikes, or hypothetical proteins on NCBI [66,67]. According to these authors, phages like the podovirus G7C present two RBPs and a structural organization where a longer RBP is directly connected to the phage particle by its N-terminal anchor domain, and the second and smaller one is attached to the first RBP and does not interact with the phage particle, forming an anchor-branched complex. Based on the structural and sequence similarities, this is probably the arrangement found on the acadeviruses with two putative depolymerase enzymes [66]. In the case of BigMira and MidiMira, the phage tail fiber Gp51 is the RBP that directly connects to the phage particle and anchors the hypothetical protein Gp52. The *Acadevirus* members with just one putative depolymerase enzyme presumably present the T7/K1F organization, the simplest one, where the phage tail fiber directly connects to the phage particle. Once the depolymerase halo is present in both acadeviruses having a single or double RBP organization, the phage tail fibers are most likely the proteins where the depolymerase enzyme actively cleaves the LPS. The second RBP, the hypothetical protein (Gp52), probably has a different enzymatic specificity. HHPred predicted this second protein as a phage tail spike that functions as a lyase/hydrolase, but HMMR prediction failed. The tertiary structure prediction also obtained low folding reliability in several regions, leading to the conclusion that this protein may have a structure uncommonly found in the available databases.

However, although the depolymerase halo is a characteristic of *Acadevirus*, phage MidiMira does not have it. This was one of the reasons why BigMira and MidiMira phages were considered distinct until the sequencing result was obtained. As these phages not only look similar, but also share the same genome size, the same G+C content, the same number of ORFs, and more than 99% sequence identity, only an SNP analysis was able to differentiate them and indicate their differences. The results showed that only four missense point mutations distinguish one phage from the other (Table 3), all occurring in the phage tail fiber Gp51. As previously discussed, this protein was predicted as having a depolymerase activity, being potentially responsible for the cleavage of LPS in the host cell wall [65,68]. These mutations not only led to a change of amino acids but also changed their interactions, since the characteristic of the amino acids generated by the mutated triplet of nucleotides is different from the original one [69], thereby potentially explaining the difference in halo formation.

Figure 6 illustrates the monomeric structure of the tail fiber Gp51 of BigMira and MidiMira. It was not possible to identify conformational alterations just by observing the predicted models, but three of the four mutations happened on the C-terminal region of the protein. They occur in a region related to the receptor recognition and/or protein trimerization of the phage tail fiber [66,68]. The substitution between amino acids with the most distinct characteristics happened in the beta-catalytic region of the enzyme. Tyrosine is a hydrophobic aromatic amino acid that can be involved in stacking and hydrogen bonding interactions. Cysteine is an uncharged hydrophilic amino acid, capable of forming disulfide bonds with other cysteines present in the protein. In fact, disulfide bonds between cysteine residues are one of the forces that drive and stabilize proteins folding into a tertiary structure. The fact that tyrosine is a hydrophobic amino acid and cysteine is hydrophilic can lead to discrete differences in the folding of this protein, since hydrophobic groups tend to face the internal side of the protein and hydrophilic ones tend to be located on its surface. Thus, this minor alteration, strengthened by the other amino acid alterations, resulted in a change in the phage tail fiber structure and impacted its enzymatic function [69]. Mutations in the catalytic domain of an enzyme may potentially alter or inactivate the enzymatic catalytic pocket, which is a three-dimensional region, usually with a specific conformation, that contains the amino acid residues and substrates necessary for catalyzing a reaction. In the case of phage RBPs, alterations in the catalytic pocket might result in a switch of receptors, which usually incorporates extra point mutations in the C-terminal domain but still allows the phage to infect the same host [70,71].

The clinical strains of *P. mirabilis* were sequenced to find evidence that explains why BigMira and MidiMira could only infect their isolation host. Figure 7A illustrates their pan-genome composition, and Figure 7B displays the similarity between the genes just related to the LPS locus. The separation of the isolation strain *P. mirabilis* MCS into a single clade is an indication that its LPS is different from that of the other evaluated strains. Although this difference may seem discrete, phage depolymerases are extremely specific [66,68]. The pan-genome analysis indicated that the closest isolate to *P. mirabilis* MCS is *P. mirabilis* 082; however, the analysis of the LPS locus showed that they are quite different in this regard.

Finally, the prediction of antibiotic resistance genes within the genomes of the clinical isolates of *P. mirabilis* provided interesting insights for this study. *P. mirabilis* is described as naturally resistant to polymyxins and tetracyclines and susceptible to β-lactams, chloramphenicol, fluoroquinolones, and aminoglycosides [4,72,73]. Besides that, over time, more and more *P. mirabilis* strains harboring other AMR genes were isolated, such as those containing *bla*_TEM_ genes, which confer resistance to the first generation of beta-lactams (such penicillin), the mutated variant of *gyrB*, which confers resistance to fluoroquinolones, and the *aacs* and *aphs* genes, which confer resistance to aminoglycosides [4,72,73]. In fact, *bla*_TEM_ genes were found in three out of the ten *P. mirabilis* strains evaluated in this study. Three out of ten was also the number of strains containing aminoglycoside resistance genes, although in this case, they were present in up to five different variants, in the strains with the largest AMR gene profile (*P. mirabilis* MCS, 204, and 195). *gyrB* was considered a core gene for the isolates in this study, as it was present in all of them. However, besides the presence of the previous genes being alarming, the genes *bla*_CTX_ (which confers resistance to an extended spectrum of beta-lactamases (ESBLs)) and *bla*_OXA_ (which confer resistance to carbapenems) potentially impact public health and are considered by the World Health Organization (WHO) as causing increasing concern; the antibiotics these genes confer resistance to are used for difficult-to-treat infections and the increase in bacterial strains resistant to them means a significant reduction in therapeutic alternatives [15,17]. Variants of both genes are present in *P. mirabilis* MCS, together with genes that confer resistance to sulfonamides, chloramphenicol, streptogramins, diaminopyramidines, antiseptics, tetracyclines (two), and aminoglycosides (three), besides the core genes (five), totaling seventeen AMR genes in a single bacterial isolate. As all the strains evaluated in this study originated from clinical settings, the presence of resistance genes beyond those classically described for the species is not uncommon. Even so, the resistance profile found in *P. mirabilis* MCS proved to be much more worrying than expected. This isolate not only has a large number of resistance genes, but they also confer resistance to a large range of antibiotic groups. This finding indicates that antibiotic treatment of this bacterial strain is complex and that an infection induced by it has a high potential for becoming difficult to treat. Thus, whether used in tandem with antibiotics or as part of a specific phage cocktail, the BigMira and MidiMira phages would be valuable tools for combating infections caused by the isolate *Proteus mirabilis* MCS or closely related isolates.

## 5. Conclusions

The use of phages to treat bacterial infections has resurfaced as an attractive alternative, and in fact, some studies already demonstrate the effectiveness of using phage cocktails to control *P. mirabilis* infections [7,36,74,75]. Phages BigMira-UFV01 and MidiMira-UFV02 belong to the genus *Acadevirus* and have characteristics that classify them as excellent candidates for phage therapy, such as the absence of lysogeny genes, good stability, and the presence of enzymes of high biotechnological interest. Although these phages produce different plaques, only four missense point mutations differentiate one from the other. The fact that these phages were isolated against a very pathogenic strain of *P. mirabilis* shows how relevant phage therapy is and how phages have the potential to become an adjunct treatment for difficult-to-treat infections.

## Figures and Tables

**Figure 1 microorganisms-11-02141-f001:**
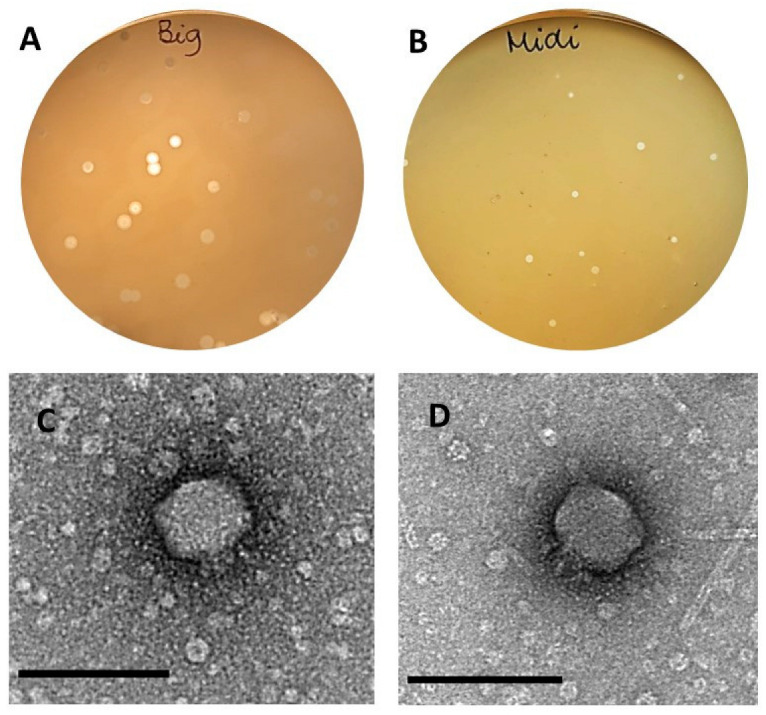
Lysis plaques and TEM morphology. (**A**) BigMira plaque morphology. The larger plaques are surrounded by halos. (**B**) MidiMira plaque morphology. The plaques are smaller compared to BigMira plaques, but still very clear through the plates, without halos, even after several days of incubation. (**C**,**D**) Transmission electronic microscopy (TEM) of BigMira and MidiMira viral particles. Both present the same podovirus morphology; bar = 100 nm.

**Figure 2 microorganisms-11-02141-f002:**
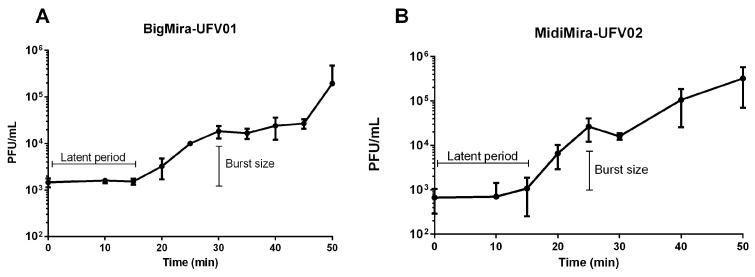
One-step growth curves. The initial phage titer was 10^3^ PFU/mL. (**A**) BigMira one-step growth curve. The calculated latent period was 15 min, and the burst size was 13 phage particles per infected cell (p.p/i.c). (**B**) MidiMira one-step growth curve. For this phage, the latent period was also approximately 15 min, and the calculated burst size was 39 phage particles per infected cell (p.p/i.c).

**Figure 3 microorganisms-11-02141-f003:**
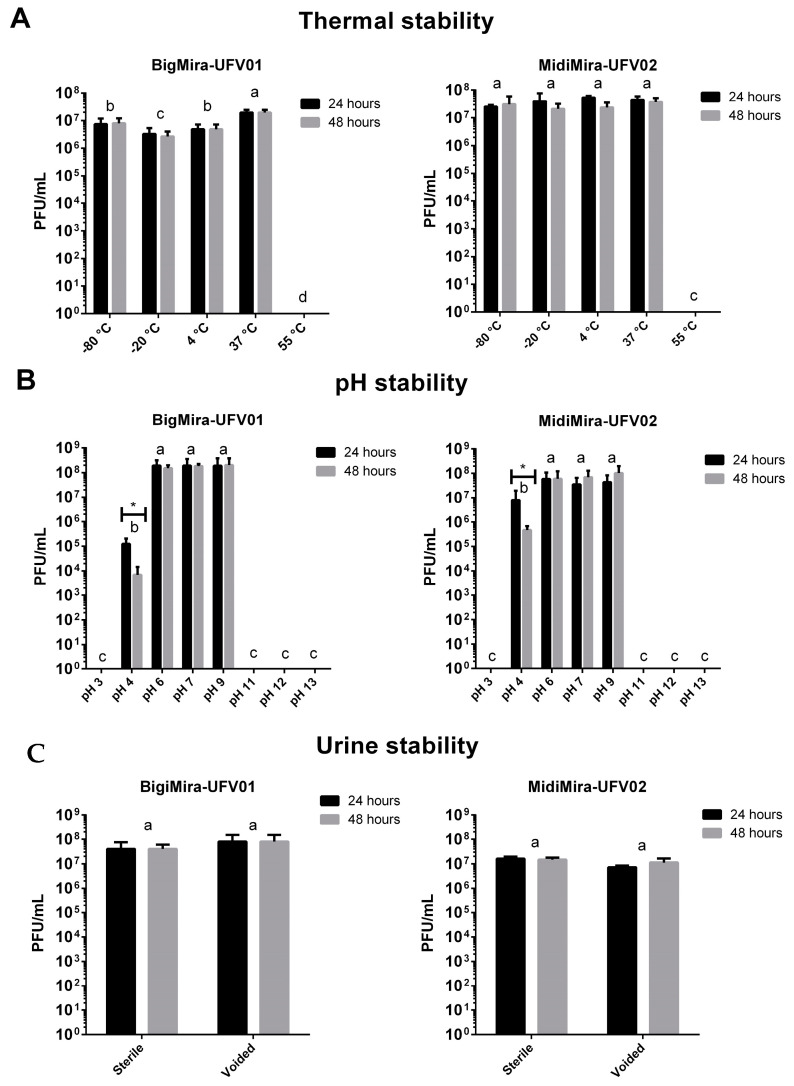
BigMira and MidiMira stability. To perform the statistical analysis, the phage titer was normalized using the Log10 baseline, but the graphics show the original titer (PFU/mL). (**A**) Phages’ thermal stability. Conditions with different letters present significant differences (*p*-value ≤ 0.05) in the phage titer. (**B**) Phages’ pH stability. (**C**) Phages’ urine stability. Conditions with different letters present significant differences (*p*-value ≤ 0.05) in the phage titer. The values with asterisks present a significant difference in phage titer based on the incubation time.

**Figure 4 microorganisms-11-02141-f004:**
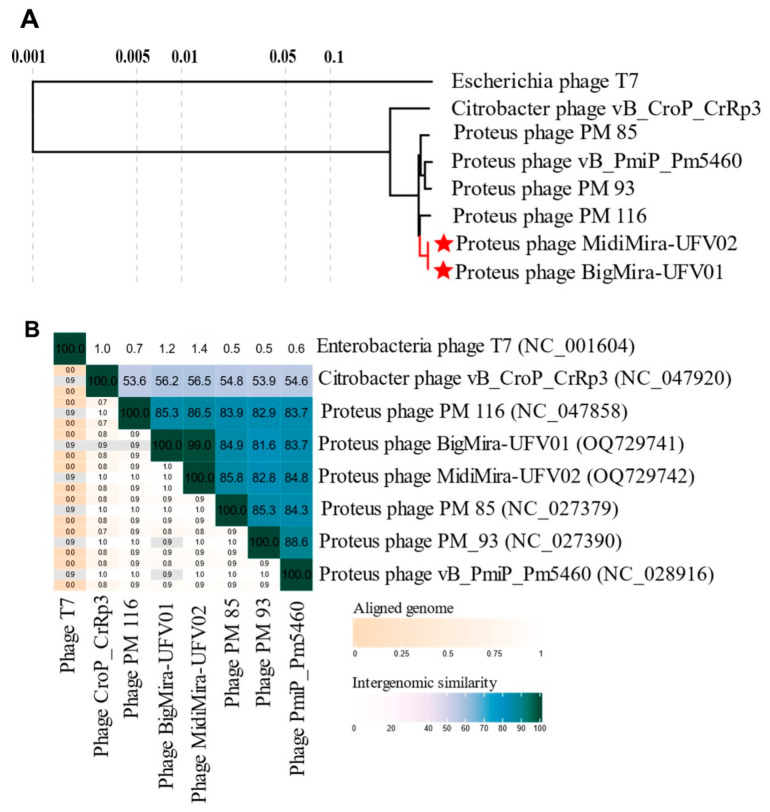
Acadevirus phylogeny. (**A**) Phylogenetic tree created only with the closest genomes (S_G_ ≥ 0.8) to BigMira and MidiMira. Acadeviruses form an independent clade. *Citrobacter* phage vB_CroP_CrRp3 was used as a related phage, but from a different genus, and the phage T7 (NC_001604) was used as an outgroup. (**B**) VIRIDIC heatmap showing the intergenomic distance among the related phages. Phages with more than 95% similarity belong to the same species, and with phages more than 70% similarity belong to the same genus.

**Figure 5 microorganisms-11-02141-f005:**
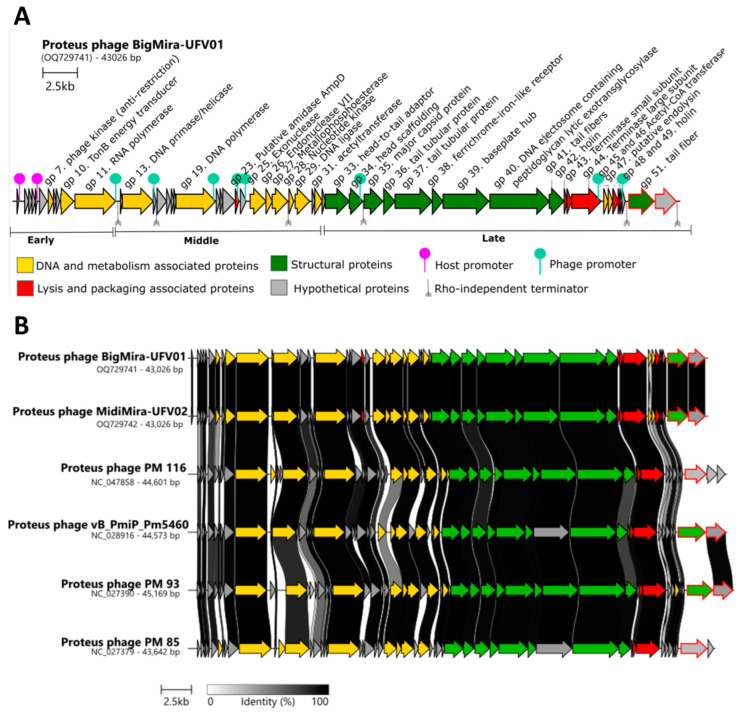
Genomic map of *Proteus* phage BigMira-UFV01 and alignment of the phages belonging to the genus *Acadevirus*. (**A**) Genomic map of *Proteus* phage BigMira. Each arrow represents an ORF, colored according to the encoded protein function: yellow—proteins associated with DNA and metabolism; green—structural proteins; red—lysis and package proteins; gray—hypothetical proteins. The ORFs contoured in red have predicted depolymerase domains. The pink pins illustrate the host-associated promoters and the green ones illustrate specific phage promoters. The Rho-independent terminators are represented by gray arrows. (**B**) Alignment of phage genomes that compose the genus *Acadevirus*. The genomes were opened using the *Proteus* phage vB_PmiP_Pm5460 as a reference. The background is colored according to the identity percentage: the blacker, the greater the identity between the ORFs. Sequences without connections do not share similarities. For the arrow colors, see the legend of Figure 5A.

**Figure 6 microorganisms-11-02141-f006:**
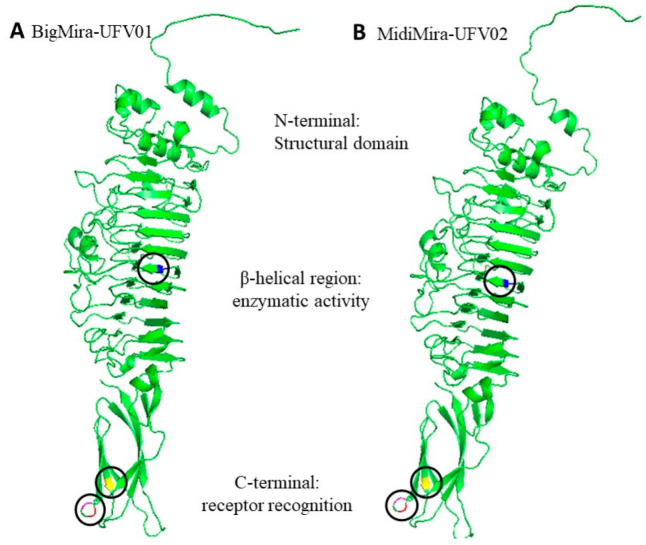
Phage tail fiber (gp 51)—monomer prediction. (**A**) BigMira-UFV01 and (**B**) MidiMira-UFV02 three-dimensional illustration of the phage tail fiber Gp52. The colored areas indicate the sites of amino acid substitution. Blue: tyrosine to cysteine (substitution A-G; nucleotide (nt) 40,548). Pink: glycine to serine (G-A; nt 41,135). Red: aspartic acid to asparagine (G-A; nt 41,141). Yellow: phenylalanine to leucine (C-A; nt 41,275).

**Figure 7 microorganisms-11-02141-f007:**
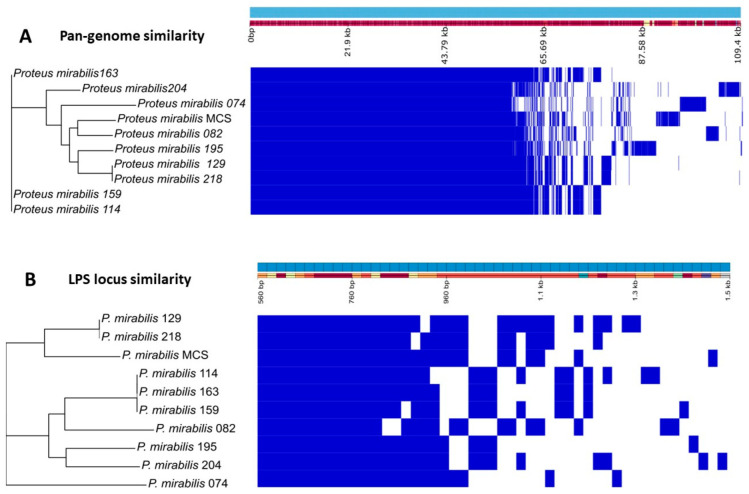
Pan-genome and LPS locus similarity of the *P. mirabilis* clinical strains.

**Table 1 microorganisms-11-02141-t001:** Host range and *Proteus mirabilis* clinical strains. The phages BigMira and MidiMira were only able to infect their isolation host strain *P. mirabilis MCS*. +: infection, −: no infection.

Host Strain	BigMira	MidiMira
*Proteus mirabilis* 074	−	−
*Proteus mirabilis* 082	−	−
*Proteus mirabilis* 114	−	−
*Proteus mirabilis* 129	−	−
*Proteus mirabilis* 159	−	−
*Proteus mirabilis* 163	−	−
*Proteus mirabilis* 195	−	−
*Proteus mirabilis* 204	−	−
*Proteus mirabilis* 218	−	−
*Proteus mirabilis* MCS	+	+
*Proteus mirabilis* 5460	−	−

**Table 2 microorganisms-11-02141-t002:** General features of the *Acadevirus* members, present within the NCBI database.

Proteus Phage	Host Range (*Proteus* spp.)	Burst Size	Genome Length (bp)	G + C Content (%)	Putative CDSs
PM 85 [56]	3/30	18	43,642	39.3	47
PM 93 [56]	2/30	75	45,169	39.4	48
PM 116 [56]	2/30	70	44,601	39.2	53
Pm 5460 [36]	16/26	46	44,573	39.6	56
BigMira	1/11	13	43,026	39.4	52
MidiMira	1/11	39	43,026	39.4	52

**Table 3 microorganisms-11-02141-t003:** Single nucleotide polymorphisms (SNPs) identified between the phages BigMira and MidiMira leading to non-synonymous mutations.

BigMira-UFV01			MidiMira-UFV02			Nucleo-Tide Position	Altered Protein
Nucleo-Tide	Amino Acid	Classification	Nucleo-Tide	Amino Acid	Classification		
A	Y (Tyrosine)	Hydrophobic aromatic	G	C (Cysteine)	Hydrophilic uncharged	40,548	Phage tail fiber (gp51)
G	G (Glycine)	Hydrophobic aliphatic	A	S (Serine)	Hydrophilic uncharged	41,135	
G	D (Aspartic acid)	Hydrophilic Acidic	A	N (Aspara-gine)	Hydrophilic uncharged	41,141	
C	F (Phenyl-alanine)	Hydrophobic aromatic	A	L (Leucine)	Hydrophobic aliphatic	41,275	

**Table 4 microorganisms-11-02141-t004:** Antimicrobial resistance (AMR) gene profile of the clinical *P. mirabilis* isolates.

Bacterial Strain	Number of AMR Genes	Gene Name	Antimicrobial Class
*P. mirabilis* MCS	12	*sulI*	sulfonamides
		*catA2*	chloramphenicol
		*vat*	streptogramins
		*dfrA1*	diaminopyramidines
		*qacEdelta1*	antiseptics
		*tetQ* *tetA*	tetracyclines
		*bla* _OXA-9_ *bla* _CTX-M-2_	β-lactams
		*aac(6′)-Iq* *aac(6′)-Ib’* *aadA*	aminoglycosides
*P. mirabilis* 204	8	*sulI*	sulfonamides
		*tetA*	tetracyclines
		*bla* _TEM-135_	β-lactams
		*aadA* *aadA2* *aph(3′)-Ia* *aph(6)-Id* *aph(3″)-Ib*	aminoglycosides
*P. mirabilis* 195	4	*catII*	chloramphenicol
		*vat*	streptogramins
		*dfrA1*	diaminopyramidines
		*aadA*	aminoglycosides
*P. mirabilis* 114 *P. mirabilis* 159 *P. mirabilis* 163	2	*vat*	streptogramins
		*dfrA1*	diaminopyramidines
*P. mirabilis* 074	1	*qnrD1*	fluoroquinolones
*P. mirabilis* 082	1	*tetQ*	tetracyclines
*P. mirabilis* 129	1	*bla* _TEM-2_	β-lactams
*P. mirabilis* 218	1	*bla* _TEM-2_	β-lactams
Core genes: *crp, kpnH*, *gyrB*, *rsmA*, and *catA4*			multidrug efflux pump, quinolones, chloramphenicol

## Data Availability

NCBI Accession numbers: *Proteus* phage BigMira-UFV01: OQ729741; *Proteus* phage MidiMira-UFV02; OQ729742; *P. mirabilis* clinical strains: BioProject ID: PRJNA979110.

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
