# Peer review of "Isolation and Characterization of the Acadevirus Members BigMira and MidiMira Infecting a Highly Pathogenic Proteus mirabilis Strain"

_microorganisms, 2023, doi:10.3390/microorganisms11092141_

Round 1
Reviewer 1 Report
This is a well-written manuscript on the isolation and characterisation of two P. mirabilis phages. Significance is in the relative scarcity of Acadeviruses of P. mirabilis, an important aetiological agent of CAUTI and bloodstream infections. The experiments were well-designed and executed, and the findings were clearly presented. Discussion points were valid but at parts could be more concise and consider relevance to therapeutic use a little more.
Introduction
Line 65 “..low production cost..”doesn’t this depend on the grade? GMP quality/medical grade phages is not low in production cost, especially when considering the need to create cocktails that have to be updated from time to time due to the emergence of phage-resistance. It may be safer to soften this a little, e.g. “…potentially low production cost..”
In the first paragraph it will be ideal to include propensity of P. mirabilis to be multi-drug resistant, since AMR is mentioned later on in the introduction, and investigated in the study.
Results
Fig 5A. Please make clearer which ORF is the predicted depolymerase. i.e. have arrow outlined in red as was done for Fig 5B.
Fig. 5B. Do the ORF colours here also match the legend for Fig 5A? If not, either include a legend for Fig 5B, or change the colours in Fig 5B to match those of 5A, and indicate the legend is for both panels.
Section 3.4. For context, please include a short sentence on why the depolymerase domain is of interest. Is there a therapeutic function, rather than a biological one?
Line 399 “…does not cluster with other clinical isolates”
Discussion
Lines 462-464, and 554-561. On the point of narrow host range of BigMira and MidiMira and phylogeny of P. mirabilis strains, I think a point to make explicit is that the strains tested are not clonal, based on pan-genomic analysis and LPS, and so the conclusion of narrow host range is justified. On this note, are there P. mirabilis phages that infect the other strains (not MCS) such that Big- & MidiMira are filling a gap in host range, albeit narrow?
Lines 473-480. On the point of phage stability parameters, state how these are relevant to the therapeutic application you’re aiming towards. i.e. UTI and pH of urine, phage product not needing a cold chain at least for 48h, etc.
Limitations should be acknowledged, such as with duration – presumably an “off the shelf” phage product is expected to be stable for more than 48h. Phages, if aiming for UTI phage therapy, should have been tested in voided and sterile urine, since it is known that phage stability in urine varies considerable depending on the phage.
Lines 493-553 would be more impactful if content could be reduced, and if relevance to therapy is integrated. There are some repetitions of results, and many words hypothesizing around predictions, simply to explain the differences between plaque morphologies and Gp51. Are SNPs in Gp51 orthologues shown to result in differences in plaque morphologies in other phages? i.e. is this a novel observation that should be highlighted?
Line 577-580. “However, besides the presence of the previous genes being 577 alarming, are the genes blaCTX (which confers resistance to an extended spectrum of beta- lactamases (ESBLs)), and blaOXA (which confer resistance to carbapenems), that potentially impacts the public health and cause increasing concern in World Health Organization 580 (WHO)[16,18].” This sentence is incomplete.
Line 584-585 “…totalizing the astonishing amount of seventeen AMR genes in a 584 single bacterial isolate.” Please rephrase this sentence, it is awkward. On this note, can the AMR genes in MCS be related back to the treatment regime of the patient from which the strain was derived? This last paragraph is a little long, given it is based on genotypic and not phenotypic data of resistance. I suggest be a more selective with content.
Reviewer 2 Report
In their paper, the authors present the isolation and characterization of two Proteus mirabilis bacteriophages, i.e. BigMira and MidiMira, infecting the highly drug-resistant P. mirabilis strain MCS. They show electron microscopy, microbiology and sequence analyses for the two phages. Moreover, they compared the P. mirabilis MCS isolation strain genome with nine other clinical isolates, all non-susceptible to BigMira and MidiMira. BigMira and MidiMira are highly similar, only differing in the plaque morphology and halo size, which the authors attribute to a few single amino acid exchanges in a putative tailspike depolymerase. HHPred analysis revealed the presence of two depolymerases gp51 and gp52. The sequence and plating data indeed suggests that the phages are O-antigen specific, however, experiments with a strain lacking O-antigen to confirm this specificity, are lacking.
Given the large number of antibiotic resistance genes in the P. mirabilis strains sequenced in this study, it is important to expand research activities on its phages with a perspective on their future therapeutic use. This paper is a classic example of a biological study describing the isolation and characterization of new bacteriophages and their hosts. As a matter of fact it does not contain a research hypothesis besides emphasizing the general need for bacteriophage-based antimicrobial treatments. This illustrates the general need for publishing experimental data as in the present study in a unified format that has the clear scope of presenting microbial diversity rather than new scientific findings. In principle, this also could be a database and the authors who are all affiliated with potent research centers and involved in efforts to implement phage therapy could go forward in this respect for the community. The present study is not new from a scientific viewpoint. Nevertheless it is important because it shows that there is a large species and bacteriophage diversity yet to be discovered.
Minor comments:
General: Why do you name gp51 as “tail fibre” whereas gp52 is a “tail spike”? Consider unifying nomenclature, also with respect to the fact that you discuss these proteins being depolymerase tailspikes rather than fibres that serve in attachment/and or cell wall puncturing.
Line 374 and following: How well does alpha fold predict protein oligomers? The training set is pdb-files of monomers, oligomers are found in experimental electron densities that alpha fold does - in my hands - not take into account.
Line 385 and following: What are the exact similarities - nearest structures - found with HHPred? Parallel beta helices with low sequence homologies may impose difficulties to alphafold below certain sequence homologies.
Line 465: “Thus, further investigation is required before this claim can be validated (Table 2).” This sentence is difficult to understand. What does “this claim” mean? The fact that Big and MidiMira are O-antigen specific phages with a narrow host range?.
Line 492: “and, therefore, are unreliable.” Why is a search tool that does not provide results that you expect “unreliable”? Rather it does not return results that you can interpret for any bacteriophage function. Remove this conclusion.
Line 507 and following: “This is probably the arrangement found on the acadeviruses with two putative depolymerase enzymes [67]. In the case of BigMira and MidiMira, the phage tail fiber Gp51 is the RBP that directly connects to the phage particle and anchors the hypothetical protein Gp52.”
You have no evidence for the arrangement that you describe here. Rephrase and clearly explain why you assume this assembly from the structural assignments made from the sequence.
Line 527 and following: “As previously discussed, this protein contains the depolymerase domain and is primarily responsible for the cleavage of LPS in the host cell wall.” Rephrase. You have no biochemical or structural evidence of the oligomeric structure nor of the location of a binding site for a receptor.
Line 536: “They occur in a region related to the receptor recognition and/or protein trimerization of the phage tail fiber.” Rephrase. You have no biochemical or structural evidence of the oligomeric structure nor of the location of a binding site for a receptor.
Line 543: Cysteine in its reduced -SH form can exist in hydrophobic environments. Tyrosine has a polar hydroxyl group. Consequently, the single amino acid exchange does not strongly change the hydrophilicity of the protein region in question. Rephrase.
Line 559: Phage depolymerases are often highly specific. Sensitivity is not appropriate as a term here, the latter would describe how small the number of hosts is that is still accessible and recognized in dilution/in a mixture/in a complex environment by the whole phage particles. Rephrase.
Reviewer 3 Report
Dear authors,
there are a lot of papers discussing roles and characteristics of phages combating bacteria medically important during experiments. However, in vitro experiments should mimic real environment.
Would you elaborate more about viral stability of BigMira and MiniMira in human urine infected by P.mirabilis strains?
Best regards,
